# Anti-Inflammatory Diets in Fertility: An Evidence Review

**DOI:** 10.3390/nu14193914

**Published:** 2022-09-21

**Authors:** Simon Alesi, Anthony Villani, Evangeline Mantzioris, Wubet Worku Takele, Stephanie Cowan, Lisa J. Moran, Aya Mousa

**Affiliations:** 1Monash Centre for Health Research and Implementation (MCHRI), School of Public Health and Preventive Medicine, Monash University, Melbourne, VIC 3168, Australia; 2School of Health and Behavioural Sciences, University of the Sunshine Coast, Sippy Downs, QLD 4556, Australia; 3Clinical and Health Sciences & Alliance for Research in Nutrition, Exercise and Activity (ARENA), University of South Australia, Adelaide, SA 5001, Australia

**Keywords:** anti-inflammatory diet, fertility, infertility, preconception, nutrition, supplementation, lifestyle, proinflammatory, review, Mediterranean diet, reproductive health and diseases

## Abstract

Infertility is a global health concern affecting 48 million couples and 186 million individuals worldwide. Infertility creates a significant economic and social burden for couples who wish to conceive and has been associated with suboptimal lifestyle factors, including poor diet and physical inactivity. Modifying preconception nutrition to better adhere with Food-Based Dietary Guidelines (FBDGs) is a non-invasive and potentially effective means for improving fertility outcomes. While several dietary patterns have been associated with fertility outcomes, the mechanistic links between diet and infertility remain unclear. A key mechanism outlined in the literature relates to the adverse effects of inflammation on fertility, potentially contributing to irregular menstrual cyclicity, implantation failure, and other negative reproductive sequelae. Therefore, dietary interventions which act to reduce inflammation may improve fertility outcomes. This review consistently shows that adherence to anti-inflammatory diets such as the Mediterranean diet (specifically, increased intake of monounsaturated and n-3 polyunsaturated fatty acids, flavonoids, and reduced intake of red and processed meat) improves fertility, assisted reproductive technology (ART) success, and sperm quality in men. Therefore, integration of anti-inflammatory dietary patterns as low-risk adjunctive fertility treatments may improve fertility partially or fully and reduce the need for prolonged or intensive pharmacological or surgical interventions.

## 1. Introduction

Infertility is defined as a failure to conceive after more than one year of unprotected intercourse in the absence of other reproductive pathologies [1]. Infertility is a global health concern, affecting 48 million couples and 186 million individuals worldwide [2]. Among all cases of infertility, 50% are attributed to female factor infertility and 20–30% to male factor infertility, while 20–30% is due to a combination of both male and female factors [3]. Infertility poses a substantial psychological, physical, and economic burden for couples trying to conceive. Current treatment options include ovarian stimulation with or without intrauterine insemination (IUI) and/or in vitro fertilisation (IVF). The high costs of IVF and other assisted reproductive technologies (ART) have made these options prohibitively expensive for many couples who wish to conceive. Whilst there are less expensive alternatives such IUI, these options are often less effective. This is mostly due to poor sperm quality leading to premature degeneration before the sperm reaches the fallopian tubes [4]. Moreover, despite the increased use of ART, the prevalence of infertility has remained stubbornly high, suggesting that other factors may be impacting on fertility and ART success. Indeed, lifestyle-related risk factors, including stress, obesity, and suboptimal diet, have been shown to exacerbate infertility [5,6,7]. These risk factors are largely modifiable, highlighting the need to identify non-intrusive and affordable strategies which can mitigate these risk factors and potentially improve fertility outcomes. 

A key modifiable risk factor is preconception diet, which has been the focus of a range of studies to date. Substantial evidence suggests that modifying preconception dietary patterns to conform with Food-based Dietary Guidelines (FBDG) is beneficial to fertility outcomes [8,9]. These dietary guidelines recommend reducing discretionary foods (typically advocating limiting intake of free sugars and foods high in saturated fats) while increasing intake of core foods typically high in unsaturated fats and components such as whole grains, vegetables, and fish [9]. Whilst the exact mechanism by which certain food groups impact upon fertility remains unknown, inflammation is thought to play a key role. Inflammation is a normal bodily process in response to infection or injury; however, prolonged sub-chronic inflammation can confer adverse effects on fertility, including disrupting menstrual cyclicity, implantation failure, endometriosis, and recurrent miscarriage [10]. Moreover, inflammation can interfere with cell trafficking pathways that are central to normal ovulatory function [10]. In men, inflammation has been shown to have a negative impact on sperm quality, a key factor underlying fecundity [11].

Dietary interventions which reduce inflammation, such as anti-inflammatory diets, in men and women during the preconception period may therefore improve fertility outcomes. However, the role of these dietary patterns in promoting fertility has not yet been established. The aim of this review is to collate the available evidence regarding the role of anti-inflammatory dietary patterns in female and male fertility, focusing on inflammation as the primary mechanism underpinning the efficacy of these interventions. We also highlight the key gaps in the literature and outline recommendations for future research in this field.

## 2. Assessing Dietary Patterns Using Diet Quality Indices

While previous research has focused on individual nutrients or single food groups, the importance of assessing dietary patterns as a determinant of overall health and disease risk has been recognised for some time [12,13]. Dietary patterns consider the totality of the diet habitually consumed by individuals and populations over a sustained period of time (months and years), rather than focusing on specific dietary constituents assessed on a single day or over the course of a few days. Dietary pattern assessments further recognise that individual foods within the diet do not function in silo; diets comprise diverse nutrients that are consumed in combination, often interacting with each other in complex ways, which may collectively impact on inflammatory status and subsequent health outcomes [12,14]. Broadly, dietary patterns can be defined as the quantities, proportions, and variety of different foods and beverages in the diet, and the frequency with which they are habitually consumed [15,16]. However, in order to attain a greater understanding of the mechanisms associated with dietary patterns in health and disease, adherence to the dietary pattern must be quantified. 

### 2.1. Quantifying Dietary Patterns

Two main methods are typically used in the research literature to quantify dietary patterns. The first method uses a priori defined numerical indices intended to capture adherence to specific dietary patterns that have been pre-defined on the basis of previous scientific evidence [16]. A priori diet quality indices are therefore used to assess adherence to specific dietary guidelines, i.e., Healthy Eating Index [17,18] or to a particular dietary pattern, such as the Mediterranean Diet (MedDiet) [19,20] or the Dietary Approaches to Stop Hypertension (DASH) diet [21]. These dietary indices quantify adherence to a dietary pattern and are thus advantageous for evaluating associations with disease risk and/or proxy health-related outcomes. Conversely, the a posteriori method is an exploratory and data-driven approach, using statistical techniques (e.g., cluster analysis, principal component analysis, etc.) to empirically derive dietary patterns collected from a specific cohort or population [16]. 

### 2.2. The Dietary Inflammatory Index

The Dietary Inflammatory Index (DII^®^) is an example of an a priori diet quality index developed to estimate the inflammatory potential of a diet [22]. On the basis of an extensive literature search aiming to quantify the overall effect of diet on inflammatory potential, a scoring algorithm using 45 pro- or anti-inflammatory food parameters was developed and has been described in detail elsewhere [22,23]. Briefly, these parameters consist of whole foods, nutrients and other bioactive compounds [22], and eleven food consumption data sets from around the world, representing a range of dietary intakes. This serves as a global reference database to provide comparative consumption data for the 45 food parameters in order to derive an individual’s DII score. Application of the DII as a relevant exposure in nutritional epidemiology provides the opportunity to explore the relationship between diet, as a determinant of non-communicable disease, and disease outcomes mediated through inflammation-related pathways [24]. Since its inception, the index has been validated by cross-sectional and longitudinal evidence showing that higher DII scores, reflective of a pro-inflammatory dietary pattern, are significantly associated with elevated plasma concentrations of proinflammatory cytokines [25,26,27,28,29]. Conversely, Moran et al. [30] developed an a posteriori diet quality index using a data-driven approach via principal component analysis, which was well-suited for assessing Mediterranean-style dietary patterns in a cohort of women with polycystic ovary syndrome (PCOS). Thus, both a priori and posteriori dietary quality indices are suited to assess diet quality in states of inflammation. 

## 3. Inflammation in Relation to Diet and Fertility

Acute inflammation is characterised by the rapid movement of plasma and leukocytes from the blood into injured tissues [14]. Conversely, chronic low-grade inflammation is characterised by levels of circulating inflammatory markers that are elevated above normal reference ranges but remain lower than in individuals with infection. A key clinical marker used to detect inflammation is C-reactive protein (CRP), an acute phase reactant, which detects inflammation and can predict risk of future diseases where inflammation is a central feature. The main distinction between acute and chronic inflammation is in the resolution of inflammation; in states of chronic low-grade inflammation there is no termination of inflammation, and persistently elevated levels of inflammatory mediators eventually have a deleterious effect on a range of body systems [15,16,17,18,19].

There is now a substantial body of evidence suggesting that various foods, nutrients, and bioactive non-nutrient plant compounds have regulatory effects on both acute and chronic inflammation [31,32,33]. For example, greater consumption of fruits and vegetables [34,35,36], whole grains [37,38], legumes [39,40], nuts [41,42] and fish [43,44] are all inversely associated with inflammation. In contrast, higher consumption of red and processed meats [45,46,47], sugar [48,49], and ultra-processed foods [50], and excessive intakes of refined carbohydrates [51] and saturated and trans fats [52,53,54,55,56] are all positively associated with a pro-inflammatory state. However, there is a lack of consistent literature on the influence of unprocessed red meat and animal protein on inflammation, with an unclear biochemical rationale to connect them. Interestingly, despite being associated with systemic inflammation [57,58], much of the available literature suggests that dairy products have neutral to beneficial effects on biomarkers of inflammation [59]. Nevertheless, individual dairy foods vary in nutritional composition, and the interplay of diverse nutrients and bioactive compounds within the dairy matrix (e.g., fat content) as well as processing techniques (e.g., fermentation) may indeed lead to differential effects on cytokine concentrations and overall inflammation.

In turn, inflammation is an increasingly recognised factor contributing to poor reproductive and fertility outcomes, termed “inflammatory infertility” [10]. Almagor et al. [60] examined the relationship between CRP and IVF success and found that the CRP ratio (embryo transfer: oocyte pick-up) was significantly lower in conception cycles compared with unsuccessful cycles, suggesting that CRP may indicate endometrial inflammatory response and optimal receptivity. Further, low-dose aspirin, which is classified as a nonsteroidal anti-inflammatory agent, increases chances of conception for women with chronic inflammation [61].

Epigenetics, which explores how the environment can affect the functions of genes (such as DNA methylation) without affecting the DNA sequence itself [62], has been identified as a potential mechanism in which diet and inflammation intersect to affect fertility. Specifically, diet-derived epigenetic modifiers have been associated with inflammation, cancer, and fertility [63,64]. Moreover, DNA methylation as well as modifications in histone post-translational modifications and miRNA can be modified by diet and mitigate inflammation in inflammatory conditions such as gout [65,66,67]. Therefore, anti-inflammatory dietary components may reduce inflammation and fertility via epigenetic modifications. Beyond this, changes in the microbiome have been shown to induce ovarian inflammation through a high-fat dietary intake, inducing imbalances in the microbiome [68]. Moreover, recent intervention studies report differences in gut microbiota composition in females with infertility compared to those without infertility [69]. Taken together, these data represent preliminary evidence for the link between obesogenic or pro-inflammatory diets, gene function, gut health, and fertility.

### 3.1. The Anti-Inflammatory Diet

One of the first versions of an anti-inflammatory diet was published in 1995 by Dr. Barry Sears in The Zone Diet, with a revised version published in 2015 [70]. Today, there are several types of anti-inflammatory diets including the Healthy Nordic Diet, the Okinawan diet, and the MedDiet, the latter of which is the most extensively studied [71]. Differences between these diets in terms of their constituents are outlined in Table 1. Whilst the Okinawan and MedDiet are similar in that they both advocate grain carbohydrates with a low glycaemic index and limited alcohol consumption, the Okinawan diet has less fat overall. Moreover, the Nordic diet is a plant-based diet that is very similar to the MedDiet, except in the method of added fat. The MedDiet is based on olive oil whereas the Nordic diet uses primarily rapeseed (canola) oil, mostly due to geographic availability [72]. Originating from cultures of ancient civilisations that developed around the Mediterranean basin, the MedDiet involves regular consumption of olive oil (main source of dietary fat), plant foods (fruits, vegetables, legumes, nuts, seeds, etc), and moderate consumption of seafood (fish, etc.) and fermented dairy, as well as limited intake of red and processed meat, sugar, and processed foods [73]. Conversely, the ketogenic diet, which consists of a low-carbohydrate, high-fat dietary pattern, has also been reported to have anti-inflammatory and antioxidant effects [74], with reported improvements in fertility hormones (testosterone, SHBG, and others) and outcomes (menstrual cyclicity, ovulation, etc) in overweight and obese women [75]. However, while emerging evidence suggests potential benefits of ketogenic diets, current data remain inconsistent, highlighting the need for further study to clarify the impacts of ketogenic diets on inflammation and fertility.

There is now a substantial body of evidence from observational and intervention studies that has identified an inverse relationship between plant-based dietary patterns (including the MedDiet) and oxidative stress and proinflammatory biomarkers [14,76,77,78,79,80]. This is perhaps not surprising given that plant-based dietary patterns contain numerous anti-inflammatory and anti-oxidative constituents which may displace pro-inflammatory nutrients within the diet. 

Specifically, monounsaturated fats, flavonoids, vitamins C and E, and polyphenols, are key constituents of anti-inflammatory diets, commonly found in plant-based foods, berries, fish, and grain carbohydrates, all of which are staples of these dietary patterns and have been linked to reduced inflammation [71]. For instance, monounsaturated fats, mainly oleic acid, are the pre-dominant fatty acids in the traditional MedDiet due to the culinary use of extra virgin olive oil (EVOO). However, the MedDiet is also a good source of the biologically active n-3 fatty acids, eicosapentaenoic acid (EPA) and docosahexaenoic acid (DHA), from fish and seafood. EPA and DHA can also be metabolically derived from their parent 18C fatty acid, alpha-linolenic acid (ALA)—found in green leafy vegetables and some nuts [81]. 

Evidence from a recent systematic review of observational studies and RCTs indicates that consumers of the MedDiet have higher tissue levels of n-3 PUFAs [82]. The role of n-3 PUFAs in ameliorating inflammatory cytokines in patients with acute and chronic diseases has been reviewed in several systematic reviews [83,84,85,86]. These reviews consistently reported that n-3 PUFAs were associated with lower inflammatory and lipid biomarkers among diabetic, cardiovascular, and immune-compromised cancer patients [83,84,85,86]. Importantly, n-3 PUFAs act as precursors to the anti-inflammatory eicosanoids, a range of biologically active downstream mediators of inflammation, including prostaglandins, prostacyclins, thromboxanes, leukotrienes, lipoxins, resolvins, protectins, and maresins, which regulate immunity, platelet aggregation, and the inflammatory response [87,88].

Flavonoids, such as quercetin, genistein, and apigenin, are biologically active polyphenolic compounds ubiquitously found in plant-based foods with antioxidant and anti-inflammatory effects (Figure 1) [89,90]. These compounds are thought to down-regulate inflammatory mediators via reducing levels of reactive oxygen species (ROS) and inhibiting key signalling pathways (nuclear factor-kappa B [NF-kB], mitogen-activated protein kinase [MAPK], etc.). There is also cross-sectional evidence to show that vitamins C and E are potent free radical scavengers and anti-oxidants and are inversely associated with inflammation [91,92]. Further, carotenoids and polyphenols both act as potent scavengers of ROS, inhibit lipid peroxidation, and modulate redox-sensitive transcription factors involved in the up-regulation of pro-inflammatory cytokines [93,94]. A recent paper identified that certain polyphenols (such as naringenin, apigenin, kaempferol, as well as others) have epigenetic actions via miRNAs [95]. In line with this, polyphenols as supplements have been identified as a potential nutraceutical to improve general health profiles, including fertility [95]. However, whilst there are many in vivo and in vitro studies in this area, very few of these studies have been conducted in humans [95]. Therefore, further research including clinical trials with human participants is warranted to obtain a more accurate assessment of this relationship.

It has also been suggested that high intakes of soluble fibre can modulate inflammatory processes in response to the production of short-chain fatty acids, in particular butyrate, which is involved in the activation of transcription factors that regulate the expression of genes encoding proinflammatory cytokines [96,97,98]. Lastly, omega-3 fatty acids found chiefly in fatty fish, such as eicosapentaenoic acid (EPA) and docosahexaenoic acid (DHA) are precursors for anti-inflammatory eicosanoids (e.g., E-series resolvins) [99,100,101,102]. They have also been shown to exert non-eicosanoid-mediated anti-inflammatory actions on cell signalling and gene expression [99,100,101]. Subgroup analysis of prospective cohort studies, including the Physicians’ Health Study (n = 405 healthy US men) and the Nurses’ Health Study (n = 1181 healthy US women) showed inverse associations between dietary intake of EPA and DHA and circulating concentrations of CRP [103,104], intercellular adhesion molecule-1 [103], vascular cell adhesion protein-1 [103], E-selectin [103], and tumour necrosis factor receptor-1 and -2 [104]. 

The specific mechanisms whereby anti-inflammatory components may directly influence fertility outcomes remain unclear. Findings from observational studies indicate (although inconsistently) that these anti-inflammatory dietary patterns attenuate pro-inflammatory markers during pregnancy and the preconception period. Some have highlighted that adherence to the MedDiet facilitates weight loss and reductions in central adiposity, potentially resulting in improved fertility, which is consistent with other studies on hypocaloric diets reporting improvements in inflammation and fertility in the short-term [105,106]. Additionally, studies have identified improvements in inflammation even among healthy populations, independent of BMI [107]. Therefore, the effects of anti-inflammatory diets and their constituents on inflammation are likely related to the diverse and synergistic relationships between the array of vitamins and minerals, fatty acids, phytochemicals, and other non-nutritive compounds (e.g., carotenoids and flavonoids) that may modulate inflammatory processes [14,108,109].

### 3.2. The Western Diet

Western dietary patterns (characterised by excessive consumption of saturated fat, refined carbohydrates, and animal proteins) are typically associated with higher levels of inflammation (Figure 1) [14,110]. As well as being energy-dense and exhibiting a high glycaemic load and hyperinsulinemic response [111], typical Western dietary patterns contain lower levels of dietary fibre, vitamins, minerals, and other plant-derived constituents [112]. This lack of substantive diet quality and diversity ubiquitous across Western societies contributes to a state of metabolic inflammation termed “metaflammation” [113,114].

Further, the introduction of Western-style dietary patterns to East Asian regions (such as Hong Kong and South Korea) is thought to be a key driver of obesity and cardiovascular disease in the region [115]. In tandem, the total fertility rate throughout East Asia has been falling precipitously over the last 50 years, and is now below the replacement rate of 2.1 [116,117]. Poor nutrition, obesity, and chronic disease could have contributed to infertility throughout the region, alongside the possible other influencing factors (cost of living, social programs, as well as others). Further large-scale studies with stratified and subgroup analyses are needed to completely assess whether the propagation of Western-style dietary patterns in East Asian regions has, at least in part, exacerbated infertility rates.

Although several mechanisms are postulated, adherence to Western dietary patterns is thought to be involved in the upregulation of several genes (Interleukin (IL)-6, IL-1β, tumour necrosis factor [TNF]) involved in pro-inflammatory pathways [113,118,119]. A recent systematic review and meta-analysis investigating the relationship between a posteriori dietary patterns and systemic inflammation in adults showed that adherence to a Western dietary pattern was positively associated with increased concentrations of CRP, leptin, and IL-6 [120]. These inflammatory mediators play an important role in complex physiological actions which regulate whole-body metabolism, including satiety, glucose disposal, fatty acid oxidation, and adipose tissue lipolysis [121]. Moreover, the consumption of energy-dense and nutrient-poor foods, such as sugar-sweetened beverages, is positively associated with body mass index (BMI) and weight gain, and is therefore an indirect pathway through which this dietary pattern promotes and maintains the chronic pro-inflammatory state common in obesity [48,122]. It has been proposed that these pro-inflammatory pathways may be related to poor fertility outcomes in both men and women.

## 4. Anti-Inflammatory Diets and Female Fertility

Preconception nutrition has been linked to fertility, and in particular to ART success [123], with limited but promising evidence suggesting that preconception dietary behaviours may improve IVF outcomes, including oocyte and embryo quality, implantation, and successfully maintaining a pregnancy to term [124,125,126,127]. Further, while healthy dietary patterns are known to be associated with better weight management, research has demonstrated the benefits of preconception diets in improving inflammation and fertility outcomes, independent of weight changes [128,129,130]. Currently available studies examining anti-inflammatory diets in relation to fertility parameters in both females (menstruation, endometriosis, and embryo quality) and males (sperm parameters) are summarised below.

### 4.1. Menstruation

Regular menstrual cyclicity is a core component of fertility, and irregular menstrual cycles (in the absence of other causes such as stress, medication, etc.) may indicate anovulation, substantially decreasing the ability to conceive [131]. Diet has been proposed as a potentially useful method for improving menstrual cyclicity, with particular interest in the MedDiet for improving menstrual regularity and pain [132].

A cross-sectional study by Onieva-Zafra [132] compared adherence to the MedDiet and consumption of local foods in 311 Spanish female university students in reference to menstrual characteristics. Approximately 55.3% of women had moderate adherence to the MedDiet and 29.6% had high adherence. Women with lower adherence had generally longer menstrual cycles than those with lower adherence, and menstrual bleeding was reduced in women who consumed olive oil daily. Further, a recent systematic review of 38 observational studies reported that increased consumption of fruits and vegetables, as a proxy measure for vitamin and mineral intake, was associated with reduced primary dysmenorrhea and menstrual pain [133].

Putative mechanisms have been described linking diet with menstruation via inflammation-related pathways. Prostaglandins (such as PGF2-α and PGE2), which are associated with inflammation, are responsible for mediating blood flow to endometrial tissue, thereby controlling local hypoxia and smooth muscle contraction and in turn supporting menstrual bleeding [134]. Anti-inflammatory constituents (omega-3 fatty acids EPA and DHA), which can be derived from the diet or metabolised from alpha-linolenic acid, can ease menstrual pain and dysmenorrhea, potentially by decreasing prostaglandin levels in the blood [134,135]. The incorporation of EPA and DHA into human immune cells is partly at the expense of arachidonic acid [136], resulting in less substrate available for synthesis of potent pro-inflammatory eicosanoids, including prostaglandin E2 (PGE2). While plausible, much of the literature describing this mechanism focuses on PGE2 in relation to colon cancer, with a paucity of evidence on menstrual disturbances.

Adhering to anti-inflammatory dietary components appears to have some benefits in relation to menstrual parameters, but it remains unclear whether improved menstrual cyclicity translates to improved fertility outcomes in this context. Since most of the studies in this area are observational, firm conclusions are precluded. This highlights the need for further well-designed, adequately powered, and appropriately controlled studies to establish the key links between diet and menstruation as a proxy measure of fertility. 

### 4.2. Endometriosis

Endometriosis is one of the most common gynaecological conditions, affecting six to twelve percent of reproductive-aged women. In this condition, endometrial tissue that normally lines the uterus is found ectopically. Endometriosis is described as an estrogen-dependent state of chronic inflammation and has implications for fertility, affecting 35 to 45% of women with infertility [137]. Inflammation in endometriosis impairs decidualisation, which is the process whereby the endometrium changes in preparation for pregnancy, reduces progesterone (a sex steroid with anti-inflammatory properties), and causes disruption in the endometrium [138]. Recently, dietary interventions, particularly those with anti-inflammatory properties, have been shown to have a beneficial effect on endometriosis and fertility outcomes through these mechanisms [139,140].

In a recent systematic review [139] of nine human and 12 animal studies examining the effectiveness of dietary interventions in the treatment of endometriosis, diets high in specific vitamins, fish oils, and mineral salts were associated with an overall reduction in symptoms, including dysmenorrhea among women with endometriosis. Specifically, EVOO, a key dietary constituent of the MedDiet, was shown to exert positive effects on endometriosis, including inflammation and pain management [141]. It is thought that these effects may occur through the actions of the component oleocanthal, which is structurally analogous to the non-steroidal anti-inflammatory agent ibuprofen [142]. Other studies have shown that supplementation with vitamins E and C also decrease markers of inflammation and oxidative stress in women with endometriosis, likely by ameliorating lipid peroxidation and promoting antioxidant and free radical-scavenging effects [143,144].

Based on these preliminary findings, it appears that interventions incorporating anti-inflammatory diet components may improve inflammation, oxidative stress, and ultimately alleviate pain and the overall symptoms and severity of endometriosis; however, their impact on pregnancy rates among women with endometriosis remains unclear [144]. Importantly, many studies in the systematic review [139] had moderate or high risk of bias, which limits confidence in the results. Further, only a small number of studies explicitly examined MedDiet adherence and endometriosis (n = one out of nine studies), whilst the remaining studies focused on other dietary patterns that included some anti-inflammatory components. It is also important to note that the literature on dietary interventions and endometriosis focuses primarily on pain parameters, and not fertility outcomes.

### 4.3. Polycystic Ovary Syndrome (PCOS)

PCOS is a common endocrine condition presenting with reproductive, metabolic, and psychological features [145]. Infertility is a prevalent feature of PCOS, with up to 75% of this population reporting reproductive problems [146]. The incurable nature of PCOS places a great burden on these women when attempting to conceive, leading to feelings of helplessness, stress, and anxiety. Thus, there is a need for effective symptomatic and component-specific relief strategies to improve the reproductive sequelae of the condition.

Low-grade inflammation is common in women with PCOS, with studies reporting that C-reactive protein (CRP) levels are typically elevated [147]. This heightened state of inflammation may contribute to infertility in this population, with anti-inflammatory diets proposed as a potential therapeutic avenue. A cross-sectional study surveying dietary intake of US women with overweight and obesity and PCOS-related infertility reported that poor dietary intake, particularly in relation to whole grains and fibre, was highly prevalent in the sample population [148]. It is thought that anti-inflammatory diets (with or without physical activity) may improve fertility as well as a range of cardiovascular and endocrine factors in PCOS through direct effects on modulating inflammation. This is supported by a prospective study in 18,555 premenopausal women indicating that promoting an anti-inflammatory diet through reducing intake of carbohydrates and overall dietary glucose load was protective against ovulatory infertility, including after adjusting for age and BMI [149]. There may also be beneficial effects of an anti-inflammatory diet on fertility via indirect relationships with decreased body weight and/or adiposity [150]. As yet, the direct versus indirect effects of an anti-inflammatory diet on fertility in PCOS have not been determined definitively.

A recent RCT investigated the effects of an anti-inflammatory dietary intervention on fertility parameters in 150 adult overweight women with PCOS. In this sample, participants were either presented with an anti-inflammatory dietary combination with physical activity alone for 12 weeks or with the addition of metformin. Improvements in menstrual cyclicity and spontaneous pregnancy were reported in the diet and physical activity group, with concomitant seven percent weight-loss, and these effects were not inferior to those observed in the metformin group [151]. This has been supported by another RCT that found a metformin-diet intervention (low glycemic index diet with ad libitum caloric intake) before and during pregnancy in 76 women with PCOS reduced miscarriage from 40% to 20% [152]. Therefore, lifestyle modification in-tandem or separate from pharmacological interventions (such as metformin) may be a similarly effective tool for improving PCOS symptoms. The exact mechanism is unknown, but it is thought that the hypocaloric and anti-inflammatory nature of the diets favour pro-conception pathways.

A Cochrane systematic review by Lim et al. [153] in 2019 was unable to include studies that assessed lifestyle interventions (diet, physical activity, behavioural, or combined treatments) on live birth, miscarriage, and menstrual regularity. In the review, most studies did not directly report clinical fertility outcomes in PCOS, but rather, endocrine and metabolic parameters that infer improved fertility. Thus, while the evidence linking anti-inflammatory diet interventions with PCOS is promising, available studies that report clinical fertility outcomes, particularly studies with long-term follow-up, remain scant.

### 4.4. Embryo Quality and Live Birth

Several studies have examined preconception diet in women undergoing ART, including intracytoplasmic sperm injection (ICSI) and IVF. In women undergoing ICSI, studies have produced inconsistent results. Hoek et al. [154] conducted a prospective cohort study of 41 couples undergoing ICSI, reporting that inadequate periconceptional maternal vegetable intake was negatively associated with embryo quality, with the effect size increasing two-fold in women with a BMI ≥ 25 kg/m^2^. An observational study [126] of 2659 embryos recovered from 269 patients undergoing ICSI cycles also reported that the consumption of whole grain cereals, vegetables, and fruits positively influenced embryo quality. Conversely, a 2021 systematic review and meta-analysis of eight prospective cohort studies which included data from 2229 women with 2067 embryo transfer cycles revealed that adherence to a dietary pattern consistent with a high intake of vegetables, fruits, wholegrain cereals, legumes, and fish (such as the MedDiet) was not significantly associated with ART (IVF with and without ICSI) outcomes, namely clinical pregnancy and live birth [155]. The PREPARE trial [156] is one of the first randomised controlled trials (RCTs) using IVF as a model to investigate the impact of a preconceptional dietary intervention on markers of embryo development. Here, six weeks of dietary supplementation consisting of a daily beverage rich in omega-3 fatty acids and vitamin D, coupled with an increased intake of olive oil, significantly altered the rate of embryo cleavage, indicating improved embryo quality among 111 couples undergoing IVF with or without ICSI [157].

Few studies have also investigated the relationship between DII scores as an indicator of the inflammatory potential of a diet and outcomes related to IVF treatment in subfertile and infertile women. In a cross-sectional study of 144 infertile women from Iran, Diba-Bagtash et al. [158] reported that DII scores were not associated with any treatment outcome parameters. These findings were corroborated by Sanderman et al. [159] in a systematic review aiming to identify female dietary patterns associated with IVF treatment outcomes. As such, Sanderman et al. [159] concluded that there was insufficient evidence to support recommending any single dietary pattern for the purpose of improving pregnancy or live birth rates in women undergoing IVF. Of note, three key methodological challenges were highlighted with respect to interpretation of the data, including inaccurate assessment of exposure (e.g., dietary intake data), possible heterogeneity in the number of previous pregnancy attempts at baseline, and the lack of adequate control for potential confounders.

Findings in relation to the MedDiet specifically have been mixed. In a prospective cohort study [128] of 244 non-obese women (aged 22–41 years) undergoing their first IVF treatment, greater adherence to a MedDiet was associated with ~2.7 times higher likelihood of clinical pregnancy and live birth. In another prospective cohort of 357 non-obese women who underwent a total of 608 ART cycles, women in the second and third quartiles of MedDiet adherence (indicating better adherence) had a higher probability of clinical pregnancy (Q2: 0.56 [95% CI: 0.47–0.64]; Q3: 0.57 [95% CI: 0.48–0.66]) and live birth (Q2: 0.47 [95% CI: 0.39–0.55]; Q3: 0.44 [95% CI: 0.36–0.53]) compared with women in the first quartile [160]. Similarly, an observational study among 700 Chinese women about to commence IVF treatment showed that greater adherence to a MedDiet was positively associated with greater embryo yield [161]. In contrast, among 474 Italian women (mean age: 36.6 years; range 27–45 years), Ricci et al. [162] reported no association between MedDiet adherence and successful IVF outcomes, including clinical pregnancy, live birth, oocyte yield, and embryo quality. A longitudinal analysis of the Rotterdam Periconceptional Cohort (Predict Study) [163] also found no significant associations between adherence to periconceptional paternal dietary patterns and embryonic growth, independent of maternal dietary patterns in spontaneous pregnancies or IVF/ICSI pregnancies.

## 5. Anti-Inflammatory Diets and Male Fertility

Abnormal sperm characteristics contribute to failed reproductive attempts and underpin 30–40% of male infertility and ~30% of subfertility cases requiring ART [164]. Oxidative stress has been shown to impact male fertility via altering the physiology of spermatozoa [165] and spermatogenesis [11]. Epidemiological studies demonstrate that men with infertility often experience chronic inflammation of the male reproductive tract, further exacerbating fertility issues [11].

### Sperm Quality

A Western-style diet is thought to increase oxidative stress through promoting weight gain and insulin resistance, which are linked with infertility and poor sperm quality [166]. Further, the increase in whole-body fat due to weight gain results in an increase in the production of pro-inflammatory cytokines and reactive oxygen species, which drive inflammation and oxidative stress. Thus, paternal nutritional interventions that aim to reduce inflammation by leveraging anti-inflammatory dietary components may be a source of improved sperm quality and consequently improved fertility outcomes. Research on the potential benefits of paternal nutrition on reproduction has mainly focused on parameters related to sperm quality. As summarised below, there is a consistent and growing body of evidence suggesting that greater adherence to an anti-inflammatory diet is positively associated with better sperm quality measures, including sperm concentration, total sperm count, sperm morphology, and sperm motility [167,168,169,170].

In a systematic review and meta-analysis of RCTs, Salas-Huetos et al. [171] reported that some dietary supplements may help to modulate male fertility. Specifically, supplementation with zinc, selenium, omega-3 fatty acids, and CoQ10 significantly increased sperm concentration and motility, with omega-3 fatty acids and CoQ10 additionally increasing total sperm count. This is unsurprising given CoQ10’s central role in the electron-transport chain whereby inhibition of the organic peroxide formation in seminal fluid may reduce sperm-cell oxidative stress [172,173]. Moreover, omega-3 fatty acids (EPA and DHA) possess anti-inflammatory and antioxidant properties, with potential influences on membrane composition [174]. Successful fertilisation of spermatozoa depends on the lipid composition of the spermatozoa membrane, which may be influenced by the concentration of omega-3 fatty acids [174]. With respect to dietary patterns, a comprehensive systematic review of observational studies by the same group [175] showed that dietary patterns similar to MedDiet or anti-inflammatory patterns that are high in fruits and vegetables, wholegrain cereals, fish, seafood, poultry, and low-fat dairy products were positively related to sperm quality. However, whilst overall sperm quality and count increased in the studies within the systematic reviews, it is unclear whether there was a return to normal clinical function.

Conversely, Western dietary patterns which include processed meat, potatoes, full-fat dairy products, coffee, alcohol, and sugar-sweetened beverages have been consistently associated with poor sperm quality and fecundability. A recent cross-sectional study by Nassan et al. [176] examined dietary patterns and testicular function in 2935 young Danish men, reporting that higher adherence to Western dietary patterns displayed overall poorer sperm quality compared to consumption of a mostly vegetarian diet. Further, a case-control study of 937 Iranian men (400 newly diagnosed infertile and 537 healthy individuals with no history of infertility) stratified dietary patterns into healthy, Western, mixed, or traditional diet categories [177]. After adjustment for confounders, men with higher (above median) adherence to a healthy dietary pattern displayed reduced risks of infertility compared to those with poor adherence (OR: 0.52 [95% CI: 0.33, 0.83]). On the other hand, men with higher adherence to Western and mixed dietary patterns were more likely to be infertile (OR: 2.66 [95% CI: 1.70, 4.17] and OR: 2.82 [95% CI: 1.74, 4.56], respectively). However, given the broad categorisation of these dietary patterns and the diversity of included foods within each category, the exact macronutrients or whole food components which contributed to these fertility outcomes cannot be determined from the available evidence. It is also important to note that diet quality should, in general, be viewed holistically, extending beyond individual components and their singular effects to encompass the broader impact of the diet as a whole.

Few studies have investigated the explicit relationship between DII scores and semen quality in males. In a cross-sectional analysis of 209 healthy male university students (aged 18–23 years) in Spain, a pro-inflammatory dietary pattern (as indicated by a positive DII score) was positively associated with total and progressive sperm motility, but had no relationship with total sperm count or morphology [178]. In contrast, in a clinic-based case-control study conducted in China, no association between DII scores and sperm motility was observed [179].

In light of the existing data, adopting a healthy diet incorporating anti-inflammatory components may have potentially beneficial effects in both women and men trying to conceive (subfertile couples). However, well-designed prospective studies and clinical trials are warranted to provide more definitive evidence in this context.

## 6. Limitations and Future Directions

Whilst current evidence for the potential utility of anti-inflammatory diets in fertility outcomes is promising, there are several limitations that must be acknowledged. First, nutritional intervention studies exhibit a vast diversity in treatment regimens, comparators, frequencies, and formulations. This heterogeneity precludes appropriate comparisons from being made between studies and makes the reported outcomes difficult to interpret.

Second, the MedDiet is a dietary pattern based on the traditional cuisines from the 1960s of Greece (strictly, Crete) and other countries that border the Mediterranean Sea. As such, a singular MedDiet does not exist. Some individuals following the MedDiet, or any other anti-inflammatory diet, may be consuming relatively more or less fruits, vegetables, or dairy, than others. This is further complicated by disparities in the diet quality indices that were used to quantify adherence to a given diet, given that the scoring systems used to quantify adherence are not homogeneous and do not always produce comparable results. As such, many of the aforementioned studies included in this review have assessed adherence to a MedDiet using the Mediterranean Diet Score (or an adaptation of this), developed by Trichopoulou et al. [19], which is dependent on the habitual dietary characteristics of the studied population and may not reflect true adherence to a MedDiet. To further complicate matters, there is added heterogeneity in other anti-inflammatory patterns (i.e., Nordic and Okinawan). As such, standardisation of MedDiet adherence tools will allow for more meaningful comparisons between studies and among diverse outcomes and sub-groups.

Third, evidence regarding female dietary patterns and female fertility outcomes, particularly in relation to IVF, relies mostly on observational study designs and early outcomes related to clinical pregnancy, with largely inconsistent results [159]. Inherently, this confers an increased risk of biases such as exposure misclassification, study-level confounding, and cohort selection, which may have influenced many of the reported results. Moreover, evidence linking diet to fertility is largely based on studies of single nutrients or individual food groups rather than overall dietary patterns [160]. Hence, although preconceptional exposure to anti-inflammatory dietary components may influence measures of fertility and ART success such as embryo quality and rates of clinical pregnancy and live birth, there is a paucity of evidence from intervention studies to clarify the validity of these associations. Therefore, further research is needed to examine anti-inflammatory diets and fertility outcomes using reliable, intervention-based research in order to inform food-based dietary guidelines.

Lastly, whilst dietary indices are useful for identifying and improving anti-inflammatory diet adherence for a specific population, at present there is a lack of consensus related to the application of specific MedDiet tools across different countries and regions [180]. This is largely due to the lack of clarity around the definition of anti-inflammatory diets, which results in marked heterogeneity in their operationalisation [181,182,183] and the need to consider anti-inflammatory-style dietary patterns when quantifying dietary adherence, particularly in non-Mediterranean or non-Nordic countries with vastly different culinary practices, cuisines, and food preparation methods [180].

Future research should focus on addressing the above limitations in order to develop high-quality nutrition intervention studies that are able to properly address these important research questions. The use of controlled studies (with standardised diets, inclusion criteria, etc.) with appropriately powered sample sizes is critically important to ensure the reliability of the results, while epidemiological and population-based studies are important for ensuring that findings have some degree of external validity.

## 7. Communicating Novel Approaches: Evidence to Integration

Novel dietary advice such as introducing anti-inflammatory diets in a clinical context can engage curiosity, increase motivation, elicit exploratory behaviour, and promote learning [184,185]. Conversely, emerging evidence suggests that the public are not engaging with national dietary guidelines because they find traditional health and nutrition messages to be repetitive and uninteresting [186,187,188,189,190]. This is reflected in Google^TM^ trends indicating that searches for ‘diet and inflammation’ and ‘anti-inflammatory diet’ have steadily increased over the past 10 years, rising by 44% and 67%, respectively (from December 2010 to November 2020). In comparison, searches for ‘national dietary guidelines’ or country-specific resources such as the ‘Dietary Guidelines for Americans’ or the ‘Australian Guide to Healthy Eating’ peaked in 2004–2006 and then steadily declined over the subsequent five-year period. As the effectiveness of nutrition education depends upon relevance to the individual [191], and generic nutrition messages are often disregarded because they fail to resonate with the public, it is possible that the use of novel nutrition messaging around diet and inflammation may help to improve uptake and adherence to lifestyle change in fertility treatment. Such approaches may, for instance, include describing the links between diet, inflammation, and fertility, with more complex concepts communicated to those with higher health literacy to provide an improved understanding without underestimating the patients’ ability to understand and synthesize nutrition information [192,193]. Dietary advice provided to patients should not always be short, simple, and without nutrition jargon [194], but rather should be tailored and match their current levels of understanding. In doing so, health professionals can responsibly disseminate important nutrition messages and improve public knowledge around new topics in nutrition science, such as the emerging potential benefits of the anti-inflammatory diet for fertility, as described herein.

## 8. Conclusions

In this review of the literature, we have highlighted that, despite some inconsistencies in the literature, adherence to an anti-inflammatory dietary pattern is generally associated with improved female (menstrual cyclicity, endometriosis-related measures, embryo quality, and live birth) and male (sperm quality) fertility-related outcomes, which are thought to occur through mediation of anti-inflammatory pathways. Whilst the current evidence is not sufficiently designed to allow for controlled analysis, following a healthy dietary pattern has no risk involved with a possible plethora of perceived benefits. Further, integration of these dietary patterns as low-risk adjunctive therapies for fertility could improve the efficacy of concomitant interventions or reduce the need for more invasive or unwarranted surgical or pharmacological interventions. Whilst diet is unlikely to remove the need for ART, it offers an easily implementable and low-risk option to assist men and women toward achieving their desired fertility outcomes. 

## Figures and Tables

**Figure 1 nutrients-14-03914-f001:**
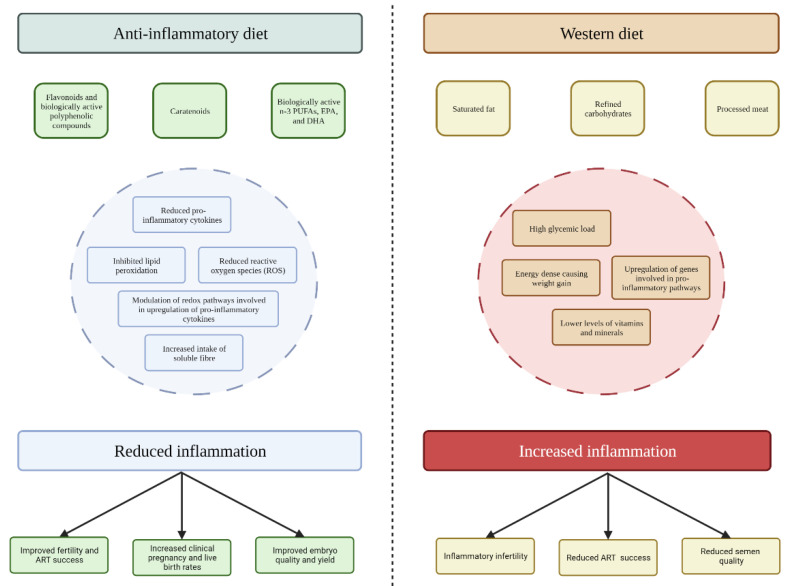
Proposed mechanisms for the impact of anti-inflammatory and Western diets on inflammation and fertility outcomes.

**Table 1 nutrients-14-03914-t001:** Dietary constituents of the Mediterranean, Nordic, and Okinawan diets.

Diet	Eat/Drink Often	Eat/Drink in Moderation	Eat/Drink Rarely	Do Not Eat/Drink
Mediterranean	Vegetables, fruits, cereals and grains, nuts, seeds, low-fat, dairy, olive oil, and low-fat dairy	White (fish or chicken) and red meat, eggs, potatoes, and wine	High-fat foods or high-sugar feeds	
Nordic	Vegetables, fruits, whole grains, nuts, seeds, low-fat dairy, canola/rapeseed oil, low-fat dairy, potatoes, and fish and seafood	Game meats (bison, antelope, etc.), eggs, cheese, and yoghurt	Red meat	Processed or refined foods, added sugars (including sugar-sweetened beverages)
Okinawan	Vegetables, fruits, soy-based foods (tofu, miso, etc.), and grains	Fish, lean meats, and alcohol	Red meat, dairy, oils, herbs/spices, nuts, seeds, and refined carbohydrates	Processed or refined foods, all added sugars

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
