# Peer review of "Anti-Inflammatory Diets in Fertility: An Evidence Review"

_nutrients, 2022, doi:10.3390/nu14193914_

Round 1
Reviewer 1 Report
Infertility seems to be a global health concern creating a significant not well-being for couples. Lifestyle factors are crucial elements to affect the outcome of fertility the major ones being dietary habits and physical activity metrics. Preconception nutritional habits are the focus of the present manuscript and specifically the anti-inflammatory diets and their effects on different fertility parameters in females and males. The review as a whole is well and clearly written and also includes the very recent relevant citations from the original literature as well as from published systematic reviews. Inclusion of an anti-inflammatory dietary pattern to every-day family habit seems to be associated with improved female and male fertility-related outcomes. I only have a few points for further clarifications.
MAJOR COMMENTS
1. Could the authors include some words regarding epigenetic factors/changes due to anti-inflammatory diets vs fertility?
2. Fast-food is a global disaster and when this type of habit was introduced in Far-East countries there were huge changes in obesity, metabolic syndrome etc. How was this affecting fertility read-outs in this region?
3. How hypocaloric diet and anti-inflammatory diet regulate specifically (mechanistically) pro-conception pathways?
4. How omega-3-FAs and CoQ10 mechanistically effect on total sperm count?
5. Microbiome also facilitates fertility effects via nutrition. The authors shoud say some words related to this important target.
Reviewer 2 Report
The authors propose an interesting topic, which indeed correlates with fertility and dietary styles.
There are some points to clarify:
- The tools used are not so representative of reality, although at the moment they are the only ones available, it would be interesting an opinion from the authors on possible improvements
- In the text and also in the figures, the intake of red meat and animal proteins, in general, is considered inflammatory (figure 1), but this is neither proven by consistent literature nor has a biochemical rationale; a fundamental point is a possible correlation with food based on processed meat; therefore it is the latter consumption that should be considered separately
- As for the omega3, it should be emphasized that DHA and EPA have an anti-inflammatory action; it is not so for the linolenic acid, which should be converted into those mentioned above two
- The ketogenic diet was not considered, which is well established, 10.3390 / antiox8080269, having an anti-inflammatory and antioxidant effect and favorable for fertility, 10.3390 / nu9030204, also used with access in case of PCOS
- The epigenetic action of a diet rich in polyphenols could also be considered, in particular related to the typical foods of the Mediterranean deity, see for example 10.3390 / antiox10020328, particularly considering the miRNAs
Therefore, probably there is no anti-inflammatory scheme, but it should be emphasized as done in the left part of figure 1 as it should be an anti-inflammatory scheme, perhaps proposing quantities to be taken.
Round 2
Reviewer 2 Report
I think that the authors have made enough improvements to make the paper published.